# Influence of Subcellular Localization and Functional State on Protein Turnover

**DOI:** 10.3390/cells10071747

**Published:** 2021-07-10

**Authors:** Roya Yousefi, Kristina Jevdokimenko, Verena Kluever, David Pacheu-Grau, Eugenio F. Fornasiero

**Affiliations:** 1Department of Neuro- and Sensory Physiology, University Medical Center Göttingen, 37073 Göttingen, Germany; roya.yousefi@med.uni-goettingen.de (R.Y.); kristina.jevdokimenko@med.uni-goettingen.de (K.J.); verena.klver@stud.uni-goettingen.de (V.K.); 2Department of Cellular Biochemistry, University Medical Center Göttingen, 37073 Göttingen, Germany; dpacheu@unizar.es

**Keywords:** protein stability, optical analysis of protein turnover, SNAP-tag, pulse-chase, Rab5a

## Abstract

Protein homeostasis is an equilibrium of paramount importance that maintains cellular performance by preserving an efficient proteome. This equilibrium avoids the accumulation of potentially toxic proteins, which could lead to cellular stress and death. While the regulators of proteostasis are the machineries controlling protein production, folding and degradation, several other factors can influence this process. Here, we have considered two factors influencing protein turnover: the subcellular localization of a protein and its functional state. For this purpose, we used an imaging approach based on the pulse-labeling of 17 representative SNAP-tag constructs for measuring protein lifetimes. With this approach, we obtained precise measurements of protein turnover rates in several subcellular compartments. We also tested a selection of mutants modulating the function of three extensively studied proteins, the Ca^2+^ sensor calmodulin, the small GTPase Rab5a and the brain creatine kinase (CKB). Finally, we followed up on the increased lifetime observed for the constitutively active Rab5a (Q79L), and we found that its stabilization correlates with enlarged endosomes and increased interaction with membranes. Overall, our data reveal that both changes in protein localization and functional state are key modulators of protein turnover, and protein lifetime fluctuations can be considered to infer changes in cellular behavior.

## 1. Introduction

Efficient protein homeostasis (proteostasis) is essential in biological systems and its imbalance leads to abnormal protein accumulation and cellular stress, which are particularly detrimental in aging organisms and tissues rich in post-mitotic cells such as brain neurons [1]. Correct proteostasis depends on the lifetimes of proteins, and is influenced by several finely regulated processes, including mRNA transcription, translation, protein stabilization through molecular interactors and protein degradation [2,3]. Approaches to study turnover in vitro and in vivo at the whole proteome level are flourishing [4,5,6,7,8,9,10,11,12] and are helping to expand our understanding of basic processes such as proteostasis regulation and cellular homeostasis. The possibility to study proteome dynamics for several thousand proteins in parallel also allows to adopt the analysis of protein lifetime changes as an additional measure for discovery purposes [13,14], similar to what is routinely achieved with other omics measures, including mRNA and protein level measurements [15]. While defining a new dimension for the study of cellular regulation at the systems biology level, the interpretation of changes in protein lifetimes opens a series of questions that remain to be thoroughly addressed.

One of these open questions is how to exactly interpret the ‘biological meaning’ of a change in protein lifetime. By definition, in order to observe a variation in the lifetime of a protein, a change in the protein synthesis and/or protein degradation rate needs to occur [16]. This might be influenced by a plethora of mechanisms such as increased mRNA levels and protein translation efficiency, protein post-translational modifications, conformational switches exposing degradation motifs, loss of protein partners or a change in the set of proteins regulating stability such as ubiquitinating enzymes [2,3,17,18].

At the same time, general overarching mechanisms at the cellular level are able to influence the stability of several proteins in parallel. For instance, correlations have been suggested for the lifetimes of proteins belonging to the same complex, with the same organelle affiliation or located in the same subcellular region [9]. As an example, mitochondrial proteins live longer than proteins localized at the endoplasmic reticulum, and proteins that belong to structural components, such as nuclear scaffolds and cytoskeletal proteins, are generally longer-lived [4,9,19]. This idea is reinforced by the observation that biochemical properties, organelle affiliations and amino acid composition can be used for predicting protein lifetimes in vivo independently from known primary degron sequences [20,21]. This suggests that, besides protein-specific regulators of protein stability, there might be more general regulatory mechanisms influencing protein lifetime that are not yet entirely understood.

The above-mentioned organelle microenvironment and subcellular location likely influence protein stability due to a number of local differences such as variation of molecular crowding [22], partitioning in protein condensates through phase separation [23], local variations of reactive oxygen species and redox state [24], and pH or salt concentrations [25]. Furthermore, even if there is a clear predetermination of how long a protein lasts due to evolutionary pressure [26], general differences in the efficiency and the nature of protein degradation pathways might play a role. This is exemplified by mitochondria where at least three main routes for protein degradation are known, including whole organelle mitophagy, mitochondrial protease-dependent degradation and mechanisms dependent on the ubiquitin-proteasome pathway [27].

Another general mechanism with a possible influence on stability involves the functionality of the protein. For example, so called ‘orphaned’ proteins are degraded faster when they fail to perform their function by not being localized to the correct compartment or inserted into the appropriate protein complex [28]. It is assumed that the functional state of a protein can influence its stability, either by directly contributing to the physical ‘wear and tear’ damage due to physical forces and molecular aging or indirectly, by modulating the protein interactors and its subcellular localization. It is assumed that mutations in a protein can modulate its functional state. Whether non-functional proteins are more prone to degradation or turned over faster than their wild-type (WT) counterparts has been a relevant issue for a long time [29].

Here, to systematically address the influence of subcellular localization and protein functional state on protein turnover, we took advantage of the widely used protein tag SNAP-tag, an O6-alkylguanine-DNA alkyl transferase that can form a specific covalent bond with O6-benzylguanine (BG) derivatives and can thus be used for fluorescence tagging and microscopy measurements [30]. First, we established and thoroughly validated an optical method for measuring protein half-life (referred to here as ‘lifetime’). Second, we fused the SNAP-tag to 10 targeting sequences and three well-characterized proteins and their respective mutants. Using these constructs, we evaluated the role of subcellular localization and the activity modulation on protein stability and lifetime. Third, after obtaining precise protein lifetimes, we followed up one serendipitous observation that the constitutively active version of Rab5a (Q79L) is longer-lived and further characterized its subcellular localization and its ability to interact with membranes.

Overall, our results indicate that both the subcellular localization and activation state are possible modulators of protein turnover.

## 2. Materials and Methods

### 2.1. Cells and Transfection

All the experiments were performed in HeLa cells certified from the German collection of microorganisms (DSMZ ACC-57), which were used within a maximum of 15 passages. Cells were cultured in high glucose Dulbecco’s modified Eagle medium (DMEM, Lonza, Basel, Switzerland)) enriched with 10% FCS (Gibco, Waltham, MA, USA), 2 mM L-glutamine (Thermo Fisher, Waltham, MA, USA), 100 U/mL penicillin and 100 mg/mL streptomycin at 37 °C, 5% CO_2_ and passaged on a regular basis.

For imaging purposes, cells were plated either on SensoPlate 96-well glass-bottom plates at a concentration of 10,000 cells per well (Greiner Bio-One, Kremsmünster, Austria) or 12-mm glass coverslips at a concentration of 100,000 cells per well. To ensure adhesion for imaging experiments, the glass was coated with 0.1 mg/mL poly-L-lysine (Sigma, St. Louis, MO, USA), rinsed with double-distilled sterile water before plating.

Protein half-life (lifetime) measurements using a SNAP-tag pulse-chase approach were adapted from previous works [20,31]. In detail, cells were transfected in solution with Lipofectamine 2000 (Invitrogen, Waltham, MA, USA) with adjustments on the manufacturer’s protocol. The DNA and Lipofectamine were diluted separately in serum-free OptiMEM medium (Gibco, Waltham, MA, USA) and incubated for 5 min. They were mixed together and incubated for 20 min. Meanwhile, cells were counted and seeded in the 96-well plate. Transfection complexes were promptly added, and the cells were allowed to attach and express the exogenous DNA for ~16 h. Before synchronized SNAP pulsing, cells were pre-incubated with 0.2 µM SNAP-Cell Block (NEB, Ipswich, MA, USA) for 1 h to block all sensors that were produced in an unsynchronized manner. For specifically pulsing a tight population of synchronized proteins, cells were incubated for 3 h with either 0.2 µM SNAP-Cell TMR (NEB, Ipswich, MA, USA) for epifluorescence imaging or SNAP-Cell 647-SiR (NEB, Ipswich, MA, USA) for STED imaging. Then, cells were chased for different time points in the presence of 0.2 µM SNAP-Cell Block, to avoid any possible staining due to binding of the residual (unbound) ligand that might be still present following the washes. To validate the localization of the membrane sensor, cells were incubated with the MemBrite pre-staining buffer (final dilution 1:1000, Membrite Fix Cell Surface Staining Kit, Biotium, Hayward, CA, USA) for 5 min and labelled with MemBrite Fix 488/515 dye diluted in medium (1:1000, Biotium, Hayward, CA, USA) for 1 h. Before fixation, cells were washed three times in warm medium and fixed afterwards with 4% paraformaldehyde (PFA, Sigma, St. Louis, MO, USA) for 30 min. After fixation, cells were quenched with 100 mM NH_4_Cl in phosphate-buffered saline (PBS) and the nuclei were stained with Hoechst 33342 (1:10,000, Thermo Fisher, Waltham, MA, USA). Adequate washing steps in PBS were performed between each step. The coverslips were embedded in Mowiol (Merck Millipore, Kenilworth, NJ, USA) and air dried overnight at room temperature. The cells in 96-well plates were kept in PBS at 4 °C until imaging. In all microscopy experiments, the cells were also co-transfected with a GFP plasmid in a 1:3 molar ratio with respect to SNAP-plasmids to ascertain that GFP-fluorescent cells also contained the SNAP constructs (see details of image analysis below).

For biochemistry, cells were electroporated using the Neon™ Transfection system (Thermo Fisher, Waltham, MA, USA) according to a previously described protocol [32]. Briefly, HeLa cells were grown to 80% confluency, harvested, washed in PBS, and resuspended in the electroporation buffer at a concentration of 5 × 10^6^ cells in 100 µL for each reaction. Plasmids were added at a final concentration of 120 µg/mL. For electroporation, two pulses were used with a width of 20 ms and voltage of 1150 V. The cells were immediately transferred to adequate DMEM medium and incubated at 37 °C with 5% CO_2_ for 48 h.

### 2.2. Constructs

All sequences used in this work are detailed in Appendix A and also summarized in Table 1. Briefly, to study the effect of subcellular location on stability of proteins, 10 constructs were designed with the addition of specific organelle sorting signals to either the N- or the C-terminus of the SNAP protein sequence. The nuclear localization signals (3× nuclear localization factor; 3×NLS) and the palmitoylation sequence were added to the N terminus of the SNAP protein sequence in order to target it to the nucleus or associate it with membranes, respectively. For localization into the endoplasmic reticulum (ER), both an N-terminal ER signal peptide and a C-terminal KDEL sequence were added to SNAP for both ER targeting and retention. Other sorting signals were fused at the C-terminus: the signal sequence of Β-galactosyl transferase and peroxisome targeting signal (PTS) for targeting to the Golgi apparatus (median and trans cisternae) and peroxisomes, respectively. Four constructs were designed to be targeted to distinct sub-mitochondrial locations (two into the mitochondrial matrix, one to the inner membrane and one to the outer membrane). The last localization construct, Lifeact, a short peptide that binds to filamentous actin (F-actin), was fused to the SNAP N-terminally. To determine the effect of activity on protein lifetime, seven constructs were designed by fusing the SNAP-tag sequence to the sequence of WT Rab5a, creatine kinase 1 and calmodulin, alongside four mutants altering the function of these proteins (see also main text for additional details). Sequences were in vitro synthetized, ordered at GenScript (Piscataway, NJ, USA) and subcloned in the pcDNA3.1^(+)^ backbone allowing expression with the CMV promoter. All plasmids were confirmed by sequencing and can be provided upon request.

### 2.3. Immunofluorescent Staining

Standard protocols for immunostaining were applied in this project. Briefly, fixed cells on glass coverslips were quenched for 15 min in 100 mM NH_4_Cl in PBS, permeabilized and blocked in staining solution containing 2% BSA (Sigma, 9048-46-8) supplemented with 0.2% TritonX-100 (Merck, Kenilworth, NJ, USA) in PBS. We used the following primary antibodies: Nogo / Reticulon 4 as an ER marker (Novus Bio, NB100-56681SS), GM130 as a Golgi marker (BD Biosciences, Franklin Lakes, NJ, USA), PMP70 as a peroxisome marker (Abcam, Cambridge, UK), TOM20 as a mitochondrial marker (Proteintech, Rosemont, IL, USA), and EEA1 as an endosomal marker (BD Biosciences, Franklin Lakes, NJ, USA). All antibodies were diluted 1:200 in staining solution (except Reticulon 4, diluted 1:400) and applied to fixed cells for 1 h at room temperature. After three 5-min washing steps with the staining solution, the following secondary antibodies were applied: goat anti-mouse STAR580 (Abberior GmbH, Göttingen, Germany), donkey anti-rabbit Alexa Fluor 488 (Biozol, Eching, Germany) or goat anti-mouse Cy3 (Dianova, Hamburg, Germany). Secondary antibodies were diluted 1:400 in staining solution and applied for 1 h. For the staining of filamentous actin, we used Alexa Fluor 488 Phalloidin (Invitrogen, Waltham, MA, USA) diluted at 1:500 in staining solution, applied similarly to secondary antibodies. Following sequential washing steps with blocking solution (2% BSA in PBS) and PBS, the coverslips were embedded in Mowiol (Merck Millipore, Kenilworth, NJ, USA) and air dried overnight at room temperature before imaging.

### 2.4. Imaging and Image Analysis

For lifetime measurement experiments, 96-well plates were imaged using a Cytation 5™ Cell Imaging Multi-Mode Reader BioTek, Winooski, VT, USA) equipped with a Sony CCD 16-bit grayscale camera. For lifetime measures, images were acquired with a 20×, 0.45 NA objective and, for each data point, 25 images were captured per well. The average of three wells was used for analysis. Image analysis was performed using custom-built macros in ImageJ [33]. Briefly, GFP positive cells were used for selecting transfected cells and SNAP-tag signal intensities were quantified within selections at different time points during chase. For additional details, see Figure 1 and the main text.

For quantification of endosome association, images were taken with an inverted Nikon Ti epifluorescence microscope (Nikon Corporation, Tokyo, Japan) equipped with a Plan Apochromat 60×, 1.4 NA oil immersion objective, an HBO-100W Lamp, and an IXON X3897 Andor camera. A minimum of five images for each biological replicate (*n* = 3) were taken. The number and size of the endosomes and the relative distribution of SNAP-Rab5aWT was calculated with customized macros in ImageJ [33]. For the analysis of the endosomal association, using ImageJ, the overall cell area was defined based on the soluble GFP signal (co-transfected with the SNAP sensors) and the average SNAP-tag signal in the whole cell was measured. Subsequently, a threshold mask was created using the signal from endosome marker EEA1, and the average SNAP-tag signal from the mask was recorded as an endosomal fraction. By subtracting the endosomal fraction from the whole-cell signal, the fraction of the cytosolic vs. endosomal association was calculated.

For the validation of construct localization, images were taken using an Abberior Instruments (GmbH, Göttingen, Germany) microscope equipped with a UPLSAPO100×, 1.4 NA oil immersion objective in confocal mode. The image size was kept constant with an *xy*-pixel size of 50 nm.

### 2.5. Western Blotting and Subcellular Fractionation

For the preparation of heavy membrane fractions, we relied on a previously established protocol [34]. Briefly, following expression of Rab5a WT and Q79L, for each replicate, two confluent 15-cm Petri dishes of cells were harvested using a cell scraper in 1 mL of ice-cold isolation buffer containing 250 mM sucrose, 100 mM HEPES (pH 7.4), 1 mM EDTA (pH 8.0), and 1 mM phenylmethylsulfonyl fluoride (PMSF), washed two times in cold PBS (4 °C). After collection, cells were homogenized by passing them 30 times though a 22G needle avoiding bubbles, centrifuged for 10 min at 1000× *g* (at 4 °C) to eliminate the pellet of unbroken cells and nuclei. The supernatant was centrifuged for 15 min at 14,000× *g* and the pellet of heavy membranes was used for further analysis.

Protein concentration was measured using a Bradford Roti^®^-Quant (Carl Roth, Karlsruhe, Germany) and similar amounts were loaded on a precast NuPage 4–12% Bis-Tris Protein gel (Thermo Fisher, Waltham, MA, USA). Western blotting on PVDF membranes (Merck, Kenilworth, NJ, USA) was performed with anti-SNAP-tag rabbit polyclonal (NEB, Ipswich, MA, USA), anti-GAPDH mouse monoclonal (Santa Cruz Biotechnology, Dallas, TX, USA), and previously described homemade rabbit polyclonal anti-LETM1 [35] antibodies using standard protocols.

### 2.6. Computation of Biochemical Parameters

Biochemical protein parameters were computed starting from the protein sequence without the region of the SNAP-tag. Biochemical properties such as the isoelectric point (pI), secondary structure elements, and the hydrophobicity (grand average of hydropathy, GRAVY) score were computed using ProtParam module in Biophyton [36].

### 2.7. Statistical Analysis

For half-life calculations, the median value of untransfected cells was subtracted from the TMR-BG intensity values of transfected cells to remove autofluorescence. Outliers were identified as exceeding three interquartile ranges (IQR) from the median value and removed from the following analyses. TMR-BG signal intensity was evaluated using one-phase decay fit and constraining plateau to greater than zero. Standard errors of the mean (SEM) for half-lives were calculated from 95% confidence interval (CI) of the half-lives of the one-phase decay fits. For analyzing the effect of cycloheximide and lactacystin on SNAP-sensors and immunoblot of GAPDH and SNAP-tag two-way analysis of variance (ANOVA) followed by Bonferroni’s multiple comparisons test was used. To compare half-lives of localization and Rab5a mutant constructs one-way ANOVA followed by Tukey’s multiple comparisons test was used. To compare half-lives of CALM1 and CKB constructs, immunoblot of Letm1 and Rab5a and STED/epifluorescence images, unpaired two-tailed Student’s *t* test was used. To analyze the correlation between half-lives and biochemical properties of the sensors, linear fit was applied and Spearman’s rank correlation coefficient *r* was used, since the data did not follow a Gaussian distribution.

Sample sizes and data presentation methods are indicated in the legends of the respective figures. The threshold for significance was set as *p* < 0.05. All statistical tests were performed using GraphPad Prism v8.3 for Windows (San Diego, CA, USA).

## 3. Results

### 3.1. SNAP-Tag Fusion Constructs for Precise Optical Measures of Protein Turnover

In this work, we designed two sets of constructs, one for evaluating the role of subcellular localization on protein turnover and the other for determining the influence of changes in functional state. All sequences were cloned in the same pcDNA3.1^(+)^ backbone plasmid driving the expression of the proteins under the control of a CMV promoter (Figure 1a, Table 1, Appendix A). To be able to pulse and chase defined protein subpopulations, we took advantage of the SNAP-tag [37], which allows labeling with a fluorescent derivative of O6-benzylguanine (for protein turnover measurements, tetramethylrhodamine-BG; TMR-BG) or to be blocked with a non-fluorescent variant named ‘SNAP-Cell Block’ (Figure 1b). Since the SNAP ligands are covalently bound to the enzyme, this tag allows to efficiently follow protein replacement over time (Figure 1c; [20]). The SNAP-tag was fused either at the N- or C-terminus of the desired targeting sequence as previously described in previous works [38,39,40,41,42,43,44,45,46,47] and as also summarized in Table 1. For evaluating the effects on the modulation of protein activity, we used three proteins and their respective mutants: (i) Rab5a and its two activity-modulating mutants [47], Rab5a-S34N (GTP-binding defective, dominant negative) and Rab5a-Q79L (GTPase defective, constitutively active); (ii) calmodulin (Calm), a Ca^2+^-binding protein with versatile interaction properties [48] and its Ca^2+^-binding defective mutant, where the aspartates of the EF-hand domains were mutated to histidines abolishing Ca^2+^ binding (4xDH; [45]); (iii) creatine kinase, an essential enzyme in ATP homeostasis and its kinase-dead mutant (CKB-C283S; [46]). To evaluate the correct localization and avoid that fusion proteins would cause mistargeting or abnormal aggregation, each plasmid was separately transfected in HeLa cells, and the distribution pattern of each protein was evaluated in live cells following a 30-min pulse with TMR-BG (Figure 1d) and also confirmed by immunofluorescence with the respective markers (Appendix A). For all proteins, the correct localization was observed.

After confirming the correct expression pattern for all constructs, we optimized a pulse and chase workflow for measuring the protein turnover of a time synchronized population of proteins (Figure 1e). In detail, cells were transfected in solution and plated in 96-well glass bottom plates compatible with high-content microscopy imaging. The dynamics of exogenous protein expression after transfection is not linear and it is characterized by an initial exponential phase followed by a plateau [49]. Since differences in protein production rates influence the measurements of protein turnover [9], we avoided the initial time following transfection and realized that a more reliable phase of protein production is reached after 16 h for cells in our experimental conditions. Thus, before pulsing, we waited 16 h after transfection. Subsequently, all SNAP-fusion constructs were blocked for 1 h, to avoid the influence of proteins expressed asynchronously. At this point, cells were pulsed for 3 h with TMR-BG, and chased for five time points (0–32 h), corresponding to a temporal range that is compatible with all analyzed proteins.

**Table 1 cells-10-01747-t001:** SNAP-tag fusion constructs used in this work. For additional sequence details, see Appendix A.

N	Description	Construct Name	SNAP Fusion	Protein of Origin	Reference
1	Nucleus (NLS)	SNAP-NLS	N-terminal	SV40	[38]
2	Endoplasmic reticulum (ER)	ER-SNAP-KDEL	C-terminal + KDEL	CALR	[39]
3	Golgi apparatus	Golgi-SNAP	C-terminal	B4GALT1	[39]
4	Actin (Lifeact)	Lifeact-SNAP	C-terminal	Abp140	[40]
5	Peroxisome	SNAP-PTS	C-terminal	LYKSRL peptide	[39]
6	Mitochondrial outer membrane	SNAP-MITO-outer	C-terminal	MAVS	[41]
7	Mitochondrial inner membrane	MITO-inner-SNAP	C-terminal	COX6a	[39]
8	Mitochondrial matrix (a)	MITO-Matrix-I-SNAP	C-terminal	COX8	[50]
9	Mitochondrial matrix (b)	MITO-Matrix-II-SNAP	C-terminal	SU9	[43]
10	Membrane	Palmitoyl-SNAP	N-terminal	ARHGEF25 (p63)	[44]
11	Wild-type calmodulin	SNAP-CaM-WT	N-terminal	CALM1.1	[45]
12	Inactive calmodulin	SNAP-CaM-4XDH	N-terminal	CALM1.1	[45]
13	Wild-type creatine kinase	CKB-SNAP-WT	C-terminal	CKB	[46]
14	Kinase-dead creatine kinase	CKB-SNAP-C283S	C-terminal	CKB	[46]
15	Wild-type Rab5a	SNAP-RAB5A-WT	N-terminal	RAB5a.1	[47]
16	Dominant negative Rab5a	SNAP-RAB5A-S34N	N-terminal	RAB5a.1	[47]
17	Constitutively active Rab5a	SNAP-RAB5A-Q79L	N-terminal	RAB5a.1	[47]

To perform image analysis and lifetime calculation in an unbiased manner (not relying on the SNAP signal) and to concentrate only on transfected cells, alongside with each SNAP construct, a GFP plasmid was co-transfected in a 1:3 molar ratio. With this excess of SNAP plasmids, we ensure that all the GFP-positive cells also contain the sensor. For this work, > 2400 images in total were quantified, requiring a robust automatized image analysis workflow (Figure 1f). We performed this by first identifying the regions of interest (ROIs) for single cells, selecting only the transfected cells (GFP positive) and finally averaging the SNAP signal intensity for several individual cells (>2500 cells per turnover sensor).

After subtracting the background, the average signal intensity per cell was normalized to time 0 h. The signal decay was plotted over time and an exponential curve was fitted for each construct. From the fitted curves, the half-lives (t_1/2_, here often referred to as lifetimes) were calculated, corresponding to the point in which the fluorescent signal reached 50% of its initial level.

### 3.2. Workflow Validation

To ensure that the workflow that we designed is reliable, we performed a series of controls. First, the efficiency and optimal concentration of the SNAP block was tested for the inner membrane mitochondrial localization, the most challenging condition since limited membrane permeability of the blocker might decrease the efficiency of this approach. HeLa cells expressing the mitochondrial inner membrane construct were blocked for 1 h with different concentrations of the SNAP-Cell Block and then pulsed for 10 min with TMR-BG. The average TMR-BG intensity decreased dramatically compared to the control even in low concentrations of SNAP-Cell Block (Figure 2a). Increasing the concentration of the blocker above 0.2 µM did not decrease the signal further, so we decided to use a final concentration of 0.2 µM of blocker for further experiments (Figure 2b). The ability of the blocker to permeate the cell membrane, the outer and the inner mitochondrial membrane indicates that the blocker is effective regardless of the challenging subcellular location.

Second, to investigate whether the fluorescent signal after the pulse represented only the proteins produced in the synchronized period, after 16 h of transfection, we blocked the asynchronous proteins with the SNAP-Cell Block while also blocking protein synthesis using cycloheximide (CHX), which interferes with translocation of the ribosome and therefore blocks translation elongation [51]. CHX was also applied during the 3 h pulse with SNAP-Cell TMR-BG for three representative constructs with distinct localization (nucleus, mitochondrial inner membrane and actin). We decided to use pulses of 3 h because they provided an optimal tradeoff between sufficient fluorescence signal for lifetime estimations and synchronization of the protein population analyzed. We also noticed that, within this time frame, the labeled sensors were correctly targeted to their respective final location (see t = 0 in the figures below). Cells treated with CHX in all three conditions show a highly significant decrease in the fluorescent signal (Figure 2c,d). These experiments confirm that, following SNAP-Cell Block and SNAP-Cell TMR-BG pulse, the labeled proteins are a synchronized population that are produced during the 3 h pulse.

Third, since we expected that the main cause of decreased fluorescent signal in transfected cells observed over time ought to be due to protein degradation, we treated the cells with lactacystin (LCY), a proteasome blocker, during the chase. We found a clear difference in signal intensity between LCY-treated and -untreated ‘control’ cells after four-hour chase of both the nucleus- and the ER-targeted SNAP constructs (Figure 2e). However, the protein targeted to mitochondrial inner membrane seems not to be affected by the LCY treatment. This was confirmed with the prolonged half-lives that were calculated in the nuclear and ER sensors upon blocking of the proteasome activity (changing from ~2.7 to ~9.8 h in the nucleus construct and from ~5.4 h to ~9.4 h in the ER-targeted construct).

In contrast, no significant change was observed for the construct targeted to the inner mitochondrial membrane (from ~4.5 h to ~4.2 h; Figure 2f), since inner membrane proteins are probably mostly degraded through alternative routes, such as AAA proteases [52]. Overall, these three control experiments demonstrated that the workflow that we established reliably reflects the turnover of a synchronized protein population and it can thus be used to measure protein turnover.

### 3.3. Influence of Subcellular Localization on Protein Lifetimes

Spatial compartmentalization allows the existence of discrete microenvironments within the cells, complementing the specialized functions of each organelle. Some similarities in the turnover rate have been shown among the proteins of individual organelles [9]. However, the effect of subcellular localization on protein turnover is not entirely understood.

In an attempt to approach this question, here, we took advantage of our established workflow and measured the turnover of the SNAP-tag sensors designed to be localized in different cellular locations. By adding different well-established targeting sequences at either N or C terminus of the SNAP-tag, we analyzed the stability of the turnover sensor for the main membrane-enclosed organelles such as the ER, the Golgi apparatus, peroxisomes and the nucleus [38,39]. To achieve the analysis of protein turnover at the sub-organelle level in the mitochondrion, we also designed four mitochondrial constructs localized to different sub compartments of the mitochondrion: the matrix, the inner membrane and the outer membrane [39,41,42,43]. We finally also assayed the stability of the turnover sensor in the proximity of the actin cytoskeleton (Lifeact-SNAP [40]) and when associated to membranes with the a palmitoyl-SNAP construct [44].

Cells were transfected and underwent pulse-chase labeling and optical turnover analysis as described above (Figure 1). Fluorescence intensity was analyzed for each construct for each time point and was plotted over time (>350 images were analyzed for each curve). We observed a diversity in turnover sensors over time for different constructs targeted to the cellular locations considered in this study (Figure 3). Most turnover sensors reached a plateau at the end of the chase period, with the exception of the actin and ER sensors that have a negative slope even at the end of the chase (Figure 3b). With this method, we measured a range of half-lives from 2.3 h to 7.2 h.

In detail, the constructs localized to the nucleus with 2.7 h, the Golgi with 3.0 h, the membrane with 3.2 h, the matrix (2.3 and 3.2 h) and the outer membrane of mitochondria (3.5 h) have lifetimes below the average (4.0 h) and the constructs associated with actin (Lifeact-SNAP, 7.2 h), targeted to the endoplasmic reticulum (ER, 5.4 h), peroxisome (5.3 h) and the mitochondrial inner membrane (4.5 h) were the longest-lived with the most significant differences from the other constructs (Figure 3c).

We also observed a difference in the turnover rate of the SNAP-constructs targeted to different mitochondria sub-compartments. The two SNAP-tag constructs designed to localize in the mitochondrial matrix show a similarly fast turnover (2.3 and 3.2 h). However, the SNAP-tag, which is fused to the signal peptide and the transmembrane domain of the COX6a, expected to face the intermembrane space of mitochondria, has a significantly longer half-life (4.5 h).

In order to exclude the effects of other possible factors that could influence the turnover of proteins, such as the presence of known degradative signals in these sequences, we tested if primary degrons were present among our constructs [21], but we could not find any (Appendix A). In addition, we excluded possible general effects due to the theoretical isoelectric point (pI), three secondary structural features, and the GRAVY score for hydrophobicity, which have previously been associated to turnover changes [9,20,36,53,54]. No correlations were observed when the sequence of the SNAP-tag was considered in the analysis (Appendix A) and only minor effects were observed if the SNAP sequences were not considered (Appendix A).

Overall, we observed 18 instances where the lifetimes of the protein localization sensors were significantly different, mostly due to the prolonged lifetimes that were measured in the protein turnover sensors for actin, ER, peroxisomes and mitochondrial inner membrane (see significance summary in Figure 3c).

These findings indicate that targeting the SNAP-tag turnover sensor to different subcellular domains has an influence on protein lifetimes; thus, subcellular localization is a modulator of protein stability.

### 3.4. Influence of Protein Activation State on Protein Lifetimes

Mutations in the protein sequence might lead to either defective or increased functionality in a protein. Here, we asked whether a change in protein functional state, minimally impacting its structural stability but preferentially stabilizing a specific activity state, would modulate the lifetime of a protein. To cover a broad range of molecular activation states in the cell, we selected exemplary proteins to be studied. We decided to consider calmodulin, a well-studied protein modulated by the binding of a small abundant cellular ion (Ca^2+^), one representative kinase and a small GTPase (Rab5), which is essential for the regulation of the secretory pathway in cells (for more details on the choice of constructs, please refer to the discussion). Using our workflow, we thus measured the turnover of the three different SNAP-conjugated proteins and their respective activity-modulating mutants (Figure 4). The first protein that we studied was calmodulin and its mutant with defective Ca^2+^-binding properties [45] (Figure 4a). Interestingly, a significant difference was observed in the turnover of the mutant, where calmodulin unable to bind Ca^2+^ ions was almost three times less stable than the WT protein, with respective lifetimes of 2.4 h and 6.1 h (Figure 4b,c).

Secondly, we analyzed the lifetimes of the enzyme creatine kinase B (CKB, [46]) and its kinase-dead mutant obtained by substituting the active cysteine with a serine (C283S; Figure 4d–f). From the representative images and the plotted curves, a trend toward stabilization of the inactive mutant could be observed, although differences between their lifetimes were not significant (respectively, 4.4 h in the WT and 6.1 h in the C283S mutant).

Finally, we compared the turnover of the early endosome small GTPase Rab5a and its two previously described mutants (Figure 4g–i; [47]). From our measurements, it was clear that the Rab5a-Q79L constitutively active mutant is more stable than both Rab5a-WT and the Rab5a-S34N GTP-binding defective mutant (the half-lives of Rab5a-WT, Rab5a-S34N, and Rab5a-Q79L were, respectively, 4.9 h, 6.8 h, and 14.1 h; Figure 4i). A ~3-fold increase in the half-life of the constitutively active mutant was measured, while there was no significant change in the half-life of GTP-binding defective mutant compared to the WT protein.

Altogether, we observed decreased stability in the nonfunctional calmodulin, no change in the CKB and Rab5a non-functional mutants, and an increase in stability in the constitutively active mutant of Rab5a.

### 3.5. SNAP-Tag Rab5a-Q79L Is More Efficiently Associated to the Heavy Membrane Fraction and Reflect Both Changes in Activity and in Subcellular Localization and Membrane Interaction Properties

Intrigued by the increased lifetime of the constitutively active Rab5a-Q79L, we further analyzed this phenotype since the activity of small GTPases has been previously shown to be correlated to morphological changes in endosomes potentially interfering with protein subcellular localization [55]. Indeed, both epifluorescence imaging and stimulated emission depletion (STED) nanoscopy confirmed these previous observations (Figure 5a,d,e). We also observed that endosomes are fewer in the Rab5a-Q79L mutant (Figure 5b), pointing to a more efficient fusion of endosomes in the mutant. Moreover, when considering the signal of the Rab5a-Q79L mutant, the percentage of molecules associated to endosomal vesicles was significantly increased in the constitutively active mutant (Figure 5c).

We then asked whether the increase in the lifetime of the mutant might be correlated to its increased association to the membrane compartment. We thus performed a biochemical analysis of the membrane distribution of the mutant with respect to the WT control. Using a homogenization protocol followed by a series of centrifugations, we separated the heavy membrane fraction from the total cell lysate (Figure 5f). Western blot analysis revealed that indeed the Rab5a-Q79L mutant was enriched in the heavy membrane fraction when compared to the wildtype (Figure 5g–k).

These results indicate that not only are there enlarged endosomes formed in the presence of Rab5a-Q79L mutant, but the protein is also associated with the membrane to a greater degree, thus indicating that more efficient membrane localization could contribute to the increased lifetime of the constitutively active mutant.

## 4. Discussion

While pulse-chase experiments using the SNAP-tag have been previously suggested for studying protein dynamics [20,31], here, we optimized and thoroughly validated an experimental workflow for measuring the lifetime of a population of time-synchronized proteins.

Our protocol is compatible with high-content imaging and automated image analysis and reliably follows the covalent fluorescent labeling of newly synthesized proteins and their turnover (Figure 1 and Figure 2). This workflow can be implemented with minimal equipment compared to mass spectrometry analysis for directly comparing the lifetimes of different proteins. When we tested the effect of protein synthesis inhibition with cycloheximide, the fluorescence signal drastically dropped for all constructs (Figure 2d), indicating that we can specifically follow proteins that are produced during the pulsing period. Several protein degradation pathways coexist, so it was not surprising that, when assessing the role of the proteasome with LCY, the degradation of the nuclear and ER sensors was inhibited, while the mitochondrial inner membrane sensor was unaffected (Figure 2e,f). As the degradation pathways in mitochondria are very articulated (for a review, see [56]), it can be hypothesized that the mitochondrial inner membrane sensor is degraded by other pathways, e.g., through mitochondrial proteases or mitophagy.

With this approach, we investigated the effect of subcellular location on protein turnover for sequences that were on average only 7.1% different from each other. Interestingly, we obtained a range of half-lives between 2.3 h and 7.2 h for our turnover sensors. As an example, Lifeact, a 17-amino-acid peptide from an actin filament binding protein in *Saccharomyces cerevisiae*, which confers actin localization [40] showed the longest SNAP-sensor half-life (7.2 h; Figure 3). Overall, these results indicate that locating a protein to a different subcellular compartment changes its stability.

Additionally, this approach even allowed us to explore differences within the same organelle, as we did in mitochondria where the constructs targeted to the matrix have shorter half-lives compared to the inner membrane-associated construct (Figure 3). Several pathways are involved in the turnover of mitochondrial proteins [57]. While ubiquitin-dependent mitophagy removes portions or entire mitochondria [58], a number of proteases play a role in individual degradation of the proteins. These proteases are specific for proteins of different mitochondrial sub-compartments [56]. Selective removal of the inner membrane proteins is mainly mediated by ‘AAA-type’ proteases while damaged matrix proteins are degraded by other enzymes including the ‘Lon-type’ and the ‘Clp’ protease [59,60]. Whether the difference in the lifetime that we observed in mitochondrial sub-compartments is due to the different degradation machineries will require further experiments.

Taking advantage of the versatility of our experimental workflow, we also tested the possibility that mutations changing the activation state of proteins can also modulate their stability. As a proof of concept, we concentrated on three illustrative examples and selected two proteins and the respective mutants to study: (a) one ion-binding protein (calmodulin), (b) one kinase (CKB) and (c) one small GTPase (Rab5a). Our choice for calmodulin was driven by the fact that it has a highly conserved structure across eukaryotes [61] and is one of the most abundant proteins capable of binding Ca^2+^, which is an essential signaling molecule in several cellular processes [62]. Calmodulin also has the advantage of bearing an EF-hand motif, the calcium binding domain, which has been crystallized and coordinates calcium in a pentagonal bipyramidal configuration that is common to several calcium-binding proteins [63]. The reason for selecting a kinase (CKB) is that mammalian genomes encode for ~500 kinases and ~100 kinase pseudogenes [64]. These enzymes are one of the most studied targets for pharmacological intervention and are a very well-defined family of proteins, characterized by stereotypic catalytic domains that can be inactivated by point mutations [64,65]. In particular, CKB is one of the most abundant kinases in some organs including the brain [66], where protein turnover plays an essential role [9,67,68]. Finally, we took into consideration the small GTPase Rab5, a ubiquitous and very well-described small GTP-binding protein involved in endosomal recycling [47]. GTPases, which include Rho, Ras and Rab proteins, are ‘molecular switches’, that are activated by GTP and inactivated by its hydrolysis to GDP. This family of proteins comprises more than 150 members overall, is highly conserved across evolution and represents a ubiquitous regulatory mechanism in eukaryotic cells [69,70].

While studying these proteins, the influence of protein activation state on protein turnover, interestingly, in two out of three instances that were analyzed, was observed to have significant differences between WT and the mutants (Figure 4). One of these is calmodulin. After binding to Ca^2+^ ions, calmodulin binds to a large number of different interactors involved in Ca^2+^ signaling pathways and therefore plays a central role in signal transduction [71]. The binding site on the EF-hand motif is surrounded by negatively charged amino acids and Ca^2+^ is stabilized between these residues [72]. By mutating these amino acids, one can create Ca^2+^-binding defective mutants [73,74]. In this study, the Ca^2+^ binding affinity of the EF-hand was abolished by substitution of four aspartate residues to histidine residues (4×DH), which led to a two-fold reduction in half-life (Figure 4a–c).

There are many examples of proteins where loss-of-function mutants have an altered turnover and, in several cases, they are relevant for pathophysiology and disease [75]. For example, in some types of hemoglobinopathies, hemoglobin mutants that are unable to bind heme are degraded within minutes and cause severe anemia, while the WT hemoglobin lasts ~120 days (the lifespan of reticulocytes; [76]). Some loss-of-function mutants fail to reach the correct localization and are depleted along their sorting path in the ER. A few classical disease-causing examples of this type include missense mutations in Aquaporin-2, cystic fibrosis transmembrane conductance regulator (CFTR), and insulin receptor, which lead to nephrogenic diabetes insipidus, cystic fibrosis and diabetes mellitus, respectively. In all of these cases, rapid degradation of the mutant proteins along the ER-trafficking pathway has been reported [77,78,79]. The rapid degradation of mutant proteins is not restricted to proteins maturing through the ER [75]. For example, rapid intra-lysosomal degradation of the mutant versions of the enzyme neuraminidase causes an inherited metabolic disease, sialidosis type I [80]. Another example is the misfolded short-chain acyl-CoA dehydrogenase which is rapidly degraded inside mitochondria and can lead to short-chain fatty acid oxidation deficiency [81]. A more recent case was reported in a career protein located in the mitochondrial outer membrane. A mutation in SLC25A46 leads to fast and selective degradation of the protein through ubiquitin-proteasome pathway [82]. Finding the exact mechanisms leading to the reduction in half-life of the ca-binding defective calmodulin warrant further studies.

The quality control mechanisms are not always efficient in removing misfolded or mutated proteins [83]. Many examples have been reported in which proteins, upon mutation, become more stable and less prone to degradation. An extensively studied example is p53, a tumor suppressor protein which is altered in more than 50 percent of human malignancies. Unlike WT p53, which is rapidly degraded by the proteasome pathway, several gain-of-function mutations cause aggregate formation and prolonged p53 lifetime, which leads to carcinogenesis. Another example of a disease phenotype linked to altered protein degradation rate was reported in transient receptor potential melastatin member 4 (TRPM4), a cation channel. Several mutations in this protein are associated with cardiac pathological phenotypes. Interestingly, a loss-of function mutation in this protein was shown to have a 30 percent faster degradation rate while a gain-of-function variant was stabilized with a doubled half-life [84].

We observed a similar effect of gain-of-function mutation in the case of the small GTP-ase, Rab5a. Although the GTP-binding-defective mutant of Rab5a showed no difference in turnover with respect to the WT, the constitutively active Q79L mutant was significantly longer-lived (by approximately three times) and localized to enlarged endosomal structures (Figure 4g–i). In mature human neutrophils, besides localization to endosomal membranes, Rab5a was shown to be abundant in the cytosol of neutrophils and upon neutrophil activation, Rab5a is more efficiently targeted to membranes [85]. Considering these observations, we hypothesized that the constitutively active mutant of Rab5a is less distributed in the cytosol, and by being surrounded with interacting proteins at the endosome, it is probably less prone to cytosolic degradation than its WT counterpart. We found that Rab5a-Q79L is indeed expressed well and does not aggregate but leads to the formation of enlarged endosomes and is more associated to membranes (Figure 5). Rab5a is a member of the Ras superfamily of monomeric G-proteins that plays an array of functions in membrane trafficking [86]; thus, only a more careful analysis of its interaction partners during its activity cycle will be able to clarify the exact molecular mechanism on the basis of its increased turnover.

In conclusion, the lifetime of a protein can change depending on its subcellular relocation and/or modulation of its function and of its molecular interactors, rendering the measurement of protein lifetimes a relevant additional parameter to consider while interrogating cellular pathways.

## Figures and Tables

**Figure 1 cells-10-01747-f001:**
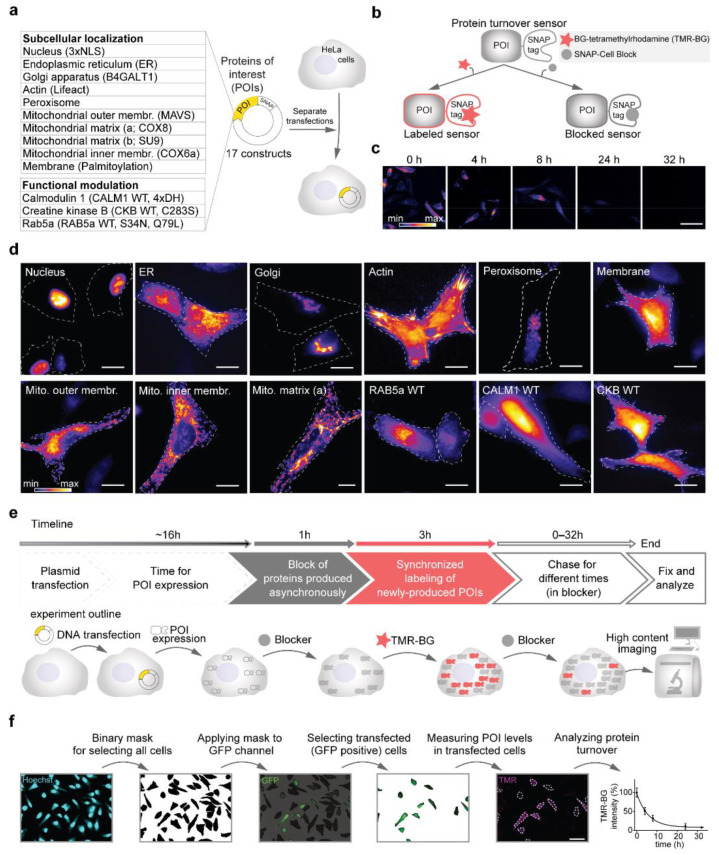
Setting up a workflow for the optical analysis of protein turnover. (**a**) SNAP-tag turnover sensors (Table 1, Appendix A) were cloned in a pcDNA3.1^(+)^ backbone. In total, 17 constructs were transfected separately into HeLa cells for measuring their turnover. (**b**) The SNAP-tags, fused to the turnover sensors, covalently bind to O6-Benzylguanine derivatives of either fluorescent dyes (such as TMR-BG in this work) or to a non-fluorescent ligand, here behaving as a ‘blocker’ (SNAP-Cell Block). (**c**) Labeled sensors can be chased over time for their turnover. (**d**) Representative images of the SNAP-tag turnover sensors expressed in HeLa cells and pulse-labeled for 30 min with TMR-BG. (**e**) Experimental pulse-chase workflow for measuring protein turnover. (**f**) Image analysis workflow. Scale bars 50 μm (**c**,**f**) and 20 μm (**d**).

**Figure 2 cells-10-01747-f002:**
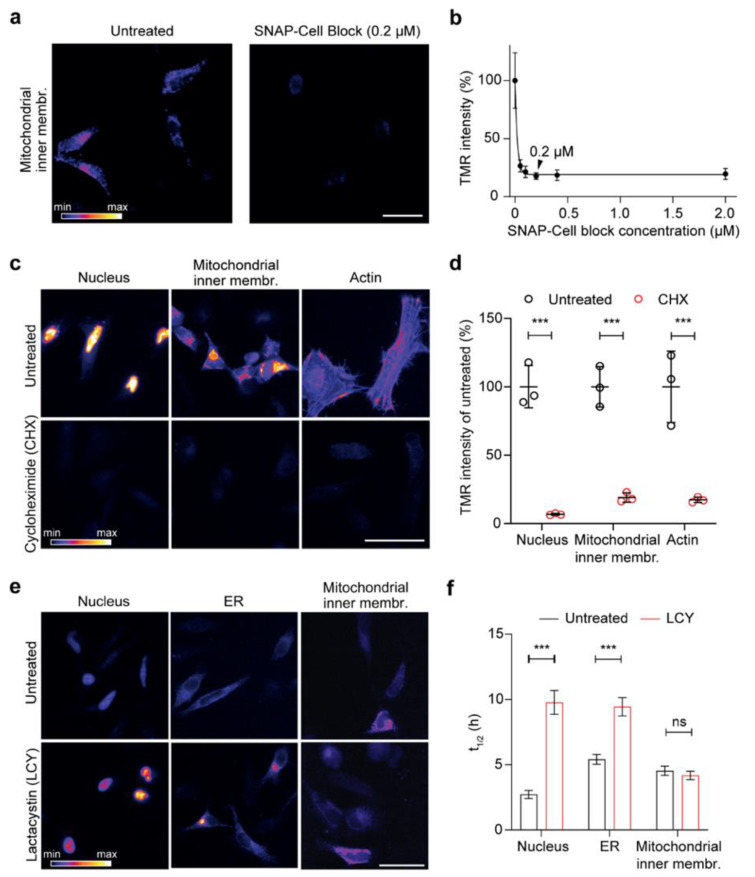
Testing the reliability of the assay for the optical analysis of protein turnover. (**a**) Representative images of cells expressing the mitochondrial inner membrane construct, either blocked with SNAP-Cell Block for 1 h or left untreated, followed by a 10 min pulse with SNAP-Cell TMR-BG. (**b**) Similar approach as in (**a**) with different concentrations of SNAP-Cell Block revealing a significant drop in TMR-BG intensity. The black arrowhead shows the selected concentration for further experiments (0.2 µM, representative image shown in (**a**). (**c**) Representative images of cells expressing NLS, mitochondrial inner membrane, and actin (Lifeact) constructs, blocked with SNAP-Cell Block for 1 h, followed by a 3-h pulse in the absence (untreated control) or the presence of cycloheximide (CHX, blocking protein synthesis). (**d**) Image intensity quantification in 75 images revealing a drop in SNAP-TMR-BG intensity after the use of CHX compared to untreated cells for all three conditions. (**e**) Block of proteasome function with lactacystin (LCY) and respective quantifications (**f**), showing that inhibiting proteasome function increases the lifetime of SNAP targeted to the nucleus or the ER, but does not affect its mitochondrial turnover. The images (**e**) correspond to the time point at 4 h after the beginning of the chase, while the quantifications in (**f**) correspond to the lifetime calculation, fitting entire chase curves as in Figure 3 (see below). Scale bars: 50 µm (**a**,**c**), 25 µm (**e**). The data are presented as mean ± 95% confidence interval (CI) (**b**) or mean ± standard error of the mean (SEM) (**d**,**f**), *n* = 3. Two-way analysis of variance (ANOVA), post hoc Bonferroni test (**d**,**f**), ns-not significant, *** *p* < 0.001.

**Figure 3 cells-10-01747-f003:**
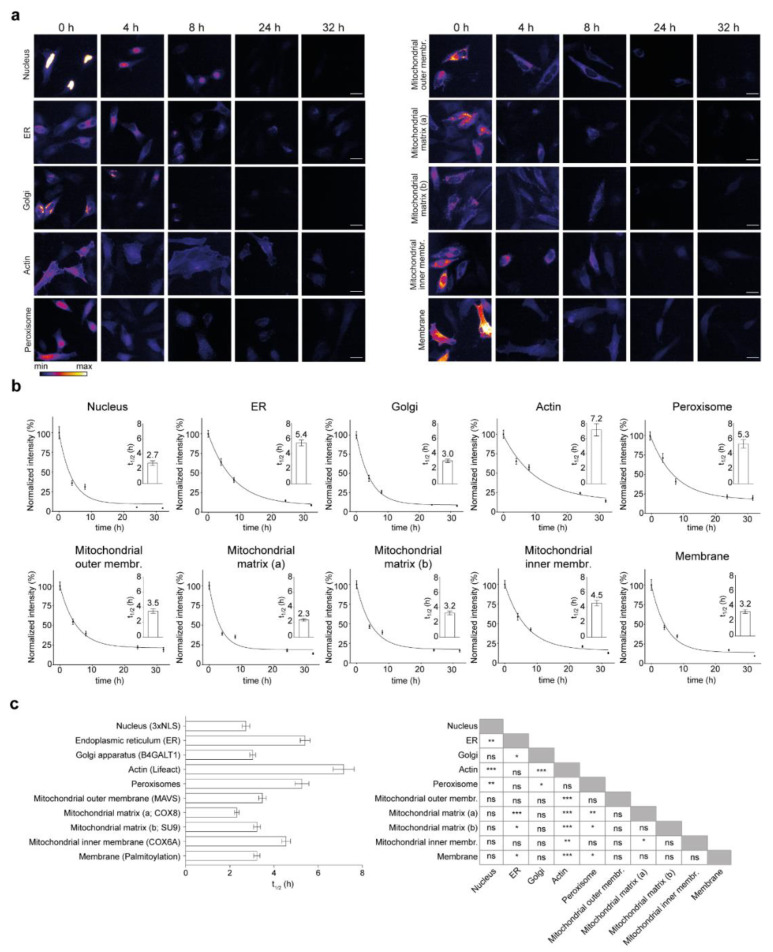
Effect of subcellular localization on lifetime of the SNAP-tag constructs. (**a**) Representative images of the fluorescence signal intensity over time in the different SNAP-tag turnover sensors targeted to the different cellular locations. (**b**) The quantification of the TMR-BG signal intensities is always normalized to the 0 h time point. Exact values of the t_1/2_ are indicated in the insets. (**c**) All half-lives of the SNAP-tagged sensors with respective comparisons. Scale bars: 25 µm. The data are presented as mean ± 95% CI (**b**) or mean ± SEM ((**b**) insets, (**c**)), *n* = 3. One-way ANOVA, post hoc Bonferroni test (**c**), ns-not significant, * *p* < 0.05, ** *p* < 0.01, *** *p* < 0.001.

**Figure 4 cells-10-01747-f004:**
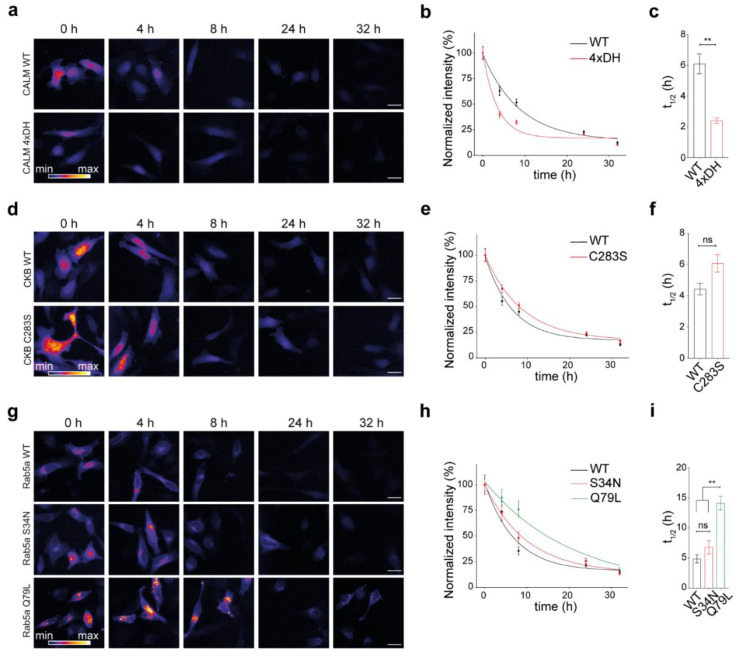
Effect of protein activity on lifetimes for three representative proteins and their respective mutants. (**a**) Representative images of the fluorescence signal intensity over time in the fusion sensor combining the SNAP-tag with either calmodulin WT (CALM WT) or the calmodulin mutant unable to bind Ca^2+^(CALM1 4xDH). (**b**) The quantification of the TMR-BG intensity over time normalized to 0 h. (**c**) Calculated half-lives for (**b**). (**d**–**f**) as panels (**a**–**c**) for the turnover sensor creatine kinase B WT (CKB WT) and its kinase-dead mutant (CKB-C283S). (**g**–**i**) as panels (**a**–**c**) for the small GTPase Rab5a (WT) and its mutants S34N (inactive), and Q79L (constitutively active). Scale bars: 25 µm (**a**,**d**,**g**). The data are presented as mean ± 95% CI (**b**,**e**,**h**) or mean ± SEM (**c**,**f**,**i**), *n* = 3. Unpaired Student’s *t*-test (**c**,**f**), one-way ANOVA followed by post hoc Tukey’s test (**i**), ns-not significant, ** *p* < 0.01.

**Figure 5 cells-10-01747-f005:**
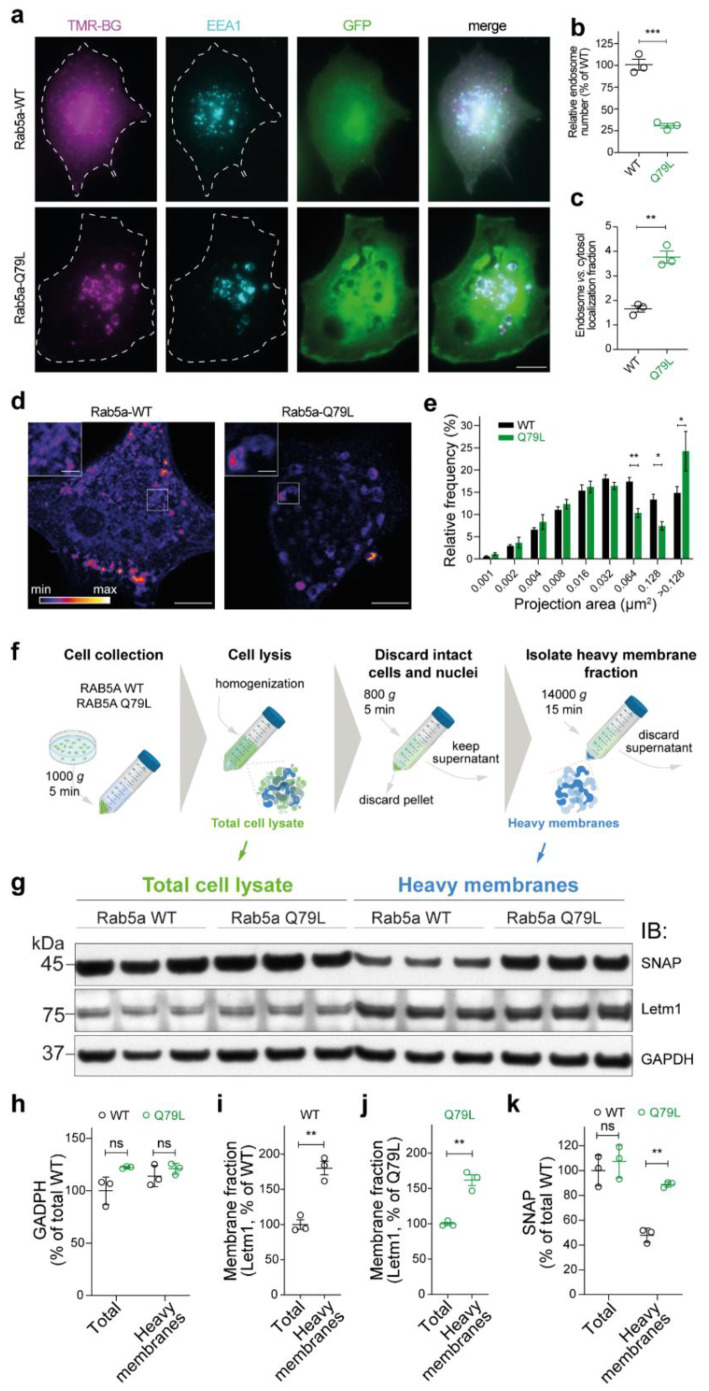
Rab5a-Q79L mutant associates more with the membrane. (**a**) Representative epifluorescence images of Rab5a-WT and Rab5a-Q79L. (**b**) Quantification of endosome number. (**c**) Quantification of endosomal vs. cytosolic distribution of the sensors. (**d**) Representative STED images of the SNAP-ligand signal for Rab5a-WT and Rab5a-Q79L lifetime sensors. (**e**). Relative frequency distribution of Rab5a-WT and Rab5a-Q79L puncta area (µm^2^) per field of view, indicating the increase in size of vesicular-like structures in the RAB5a-Q79L mutant vs. the WT. (**f**) Schematic representation of total cell lysate and heavy membrane extraction from RAB5a-WT or RAB5a-Q79L expressing cells for Western blot experiment. (P)—pellet, (S)—supernatant. (**g**) Western blot for detecting SNAP-tag (45 kDa), Letm1 (75 kDa), GAPDH (37 kDa) in the total cell lysate or heavy membrane fraction of RAB5a-WT or RAB5a-Q79L samples. The original can be found in Appendix A. (**h**–**k**) Quantitative analysis of Western blot in **g**. Scale bars: 10 µm (**a**), 5 µm (**d**) and 1 µm (in the insets in **d**). The data are presented as mean ± SEM (**b**,**c**,**e**,**h**–**k**). *n* = 3 (coverslips; b-c); *n* = 7 (cells in (**d**), WT) and *n* = 3 (cells in **d**, Q79L). *n* = 3 independent samples (**g**,**h**–**k**). Unpaired Student’s *t*-test (**b**,**c**,**i**,**j**) two-way ANOVA and post hoc Bonferroni test (**e**,**h**,**k**), ns-not significant, * *p* < 0.05, ** *p* < 0.01, *** *p* < 0.001.

## Data Availability

All original data is available upon reasonable request.

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
