# Peer review of "Influence of Subcellular Localization and Functional State on Protein Turnover"

_cells, 2021, doi:10.3390/cells10071747_

Round 1

Reviewer 1 Report

The authors have presented a manuscript entitled “Influence of subcellular localization and functional state on protein turnover”. I would like to thank you the authors for the very nicely written manuscript. The experimental design and presentation are appropriate. However few places need some clarifications.

My comments

  1. ‘For specifically pulsing a tight population of synchronized proteins, cells were incubated for 3 hours with either 0.2 µM SNAP-Cell TMR (NEB, S9106S) for epifluorescence imaging or SNAP-Cell 647-SiR (NEB, S9102S) for STED imaging. Then, cells were chased for different time points. Before fixation, cells were washed 3 times in warm medium and fixed afterwards with 4 % paraformaldehyde (PFA, Sigma, 30525-89-4) for 30 min ………”. (Line 118-122) here in the protocol description, the 2nd time blocking step has not been mentioned. The step was mentioned in the flow chart in figure 1e. Please mention in the method if blocker has been used after TMR incubation followed by chasing.
  2. Use TMR-BG in the figure for consistency as this acronym was used everywhere in the text.
  3. Did the author confirm the localization of different target proteins actually targeted to the right place? Is there any data to confirm the various subcellular targets by subcellular fractionation followed by western with marker proteins or by overlapping imaging of particular model proteins? Please provide as supplementary.
  4. For workflow validation “First, the efficiency and the optimal concentration of the SNAP block was tested. HeLa cells expressing the mitochondrial inner membrane construct…..” why specifically mitochondrial inner membrane protein was tested? Was the titration of different concentrations for both the blocker and TMR-BG tried on any other construct to ensure its universal effect? As protein localization has an influence on protein turnover, function, and activity that might vary for different constructs. Please comment.

Author Response

Reviewer #1
The authors have presented a manuscript entitled “Influence of subcellular localization and functional state on protein turnover”. I would like to thank you the authors for the very nicely written manuscript. The experimental design and presentation are appropriate.

We sincerely thank the reviewer for the overall positive assessment of our work.
However few places need some clarifications:
1. For specifically pulsing a tight population of synchronized proteins, cells were incubated for 3 hours with either 0.2 μM SNAP-Cell TMR (NEB, S9106S) for epifluorescence imaging or SNAP-Cell 647-SiR (NEB, S9102S) for STED imaging. Then, cells were chased for different time points. Before fixation, cells were washed 3 times in warm medium and fixed afterwards with 4 % paraformaldehyde (PFA, Sigma, 30525-89-4) for 30 min ………”. (Line 118-122) here in the protocol description, the 2nd time blocking step has not been mentioned. The step was mentioned in the flow chart in figure
1e. Please mention in the method if blocker has been used after TMR incubation followed by chasing.

Indeed, cells were also pulsed in the presence of blocker, we thank the reviewer for pointing this out. We have now clarified this in the method stating that, “Then, cells were chased for different time
points in the presence of 0.2 μM SNAP-Cell Block, to avoid any possible staining due to binding of the residual (unbound) ligand that might be still present following the washes.” (Page 3, lines 123-125 of the revised manuscript).
2. Use TMR-BG in the figure for consistency as this acronym was used everywhere in the text.
We thank the reviewer for the suggestion and we have amended the acronym in Fig. 1 and whenever necessary across the whole text.
3. Did the author confirm the localization of different target proteins actually targeted to the right place? Is there any data to confirm the various subcellular targets by subcellular fractionation followed by western with marker proteins or by overlapping imaging of particular model proteins?
Please provide as supplementary.

Although the targeting sequences that we employed are very well characterized and have been used in numerous other studies, we did confirm their localization with immunofluorescence localization
analysis (by overlapping imaging of particular model proteins). We have included these controls as a supplementary figure (Supp. Fig. 1 in the updated version of the manuscript).

4. For workflow validation “First, the efficiency and the optimal concentration of the SNAP block was tested. HeLa cells expressing the mitochondrial inner membrane construct…..” why specifically mitochondrial inner membrane protein was tested? Was the titration of different concentrations for
both the blocker and TMR-BG tried on any other construct to ensure its universal effect? As protein localization has an influence on protein turnover, function, and activity that might vary for different constructs. Please comment.

We decided to use the mitochondrial inner membrane for the validation of the blocker since some of the SNAP ligands (including the blocker) might have membrane permeability issues. The ability of the blocker to permeate (i) the cell membrane, (ii) the outer and (iii) the inner mitochondrial
membrane indicates that the blocker is effective even in the most challenging scenario. As also suggested by the reviewer, we have clarified this reasoning in the text (Page 8, lines 325-328).

Reviewer 2 Report

Yousefi and co-workers study the turnover of 17 SNAP-tagged protein constructs transiently expressed in HeLa cells that are targeted to different cellular compartments. Technically, the work is solid, and the authors present some interesting data, e.g. the differences in turnover rates of Rab5a wt and mutant proteins. However, the purpose of the manuscript and the reasoning behind the work are unclear. The major finding of the manuscript, as is, is that there is no clear correlation between the turnover rates of the investigated proteins. There is no hypothesis behind the work, no meaningful reasoning for selecting this set of functionally and structurally unrelated proteins, and no clear conclusions to the work. The authors should thoroughly rethink their work and re-write the manuscript in a more convincing manner or use these data in other context(s). 

Author Response

Reviewer #2

Yousefi and co-workers study the turnover of 17 SNAP-tagged protein constructs transiently expressed in HeLa cells that are targeted to different cellular compartments. Technically, the work is solid, and the authors present some interesting data, e.g. the differences in turnover rates of Rab5a
wt and mutant proteins.

We thank the reviewer for acknowledging the solidity of our work.

However, the purpose of the manuscript and the reasoning behind the work are unclear.

We apologize to the reviewer for the suboptimal representation of the results and their inappropriate discussion. As we have outlined in the letter to the Editor, protein turnover is the delicate equilibrium between protein production and degradation and changes in protein half-life (referred to as ‘lifetime’) represent variation of this equilibrium. One important open question in the field is how to interpret changes in the lifetime of proteins. This issue is relevant since nowadays protein lifetimes can be measured with other methods that we and others contributed to develop (see as an example Alevra et al., 2019). One obvious and well-accepted interpretation for the change in protein lifetimes is that the turnover of proteins depends on the protein production and degradation machineries.
While this is ultimately true, there are also other factors that might influence protein turnover, such as subcellular localization and protein functional state. In this work, we have addressed these two contributing factors and we found that they are both important determinants of protein turnover.
As an additional note, to address this, we have optimized a highly versatile workflow that can be implemented with minimum equipment (compared to mass spectrometry proteomics) for directly comparing the lifetimes of different protein sequences – although the optimization of the method was
not the main goal of our work.

The major finding of the manuscript, as is, is that there is no clear correlation between the turnover rates of the investigated proteins. There is no hypothesis behind the work.

We kindly disagree with the Reviewer. As highlighted in the text above, the hypotheses that we aimed to test are: 1) if localizing a protein to different subcellular locations can change its turnover, and/or 2) Mutating residues that are important for the specific function of a protein can influence its
turnover. We have observed that there is a clear influence of both these factors on protein turnover, as also confirmed by our stringent statistics. We believe that, when the Reviewer referred to the absence of correlation, s/he referred to our analysis of the bioinformatic parameters. This was not
the main finding of our work. As also Reviewer #3 correctly spotted, we initially left this in the main text since we “aimed at providing a complete picture of our results”. Nevertheless, since this is not
essential for our conclusions and can be misleading, we have only briefly discussed these correlations at the beginning of the results (Page 10, lines 437-445) and moved the other findings to the supplement (Page 26, Supp. Fig. 3).

[There is] no meaningful reasoning for selecting this set of functionally and structurally unrelated proteins.

We agree with the reviewer that the choice of proteins was initially not clarified. This was also the first comment of Reviewer #3. For a detailed explanation of our choice please refer to the answer to Reviewer #3 (comment #1 from Reviewer #3 in the following page). We have added the missing information about our reasoning in the main text (Page 12, lines 477-482 and Page 15, lines 581-600).

[There is] no clear conclusions to the work. The authors should thoroughly rethink their work and rewrite
the manuscript in a more convincing manner or use these data in other context(s).

We followed the suggestion of the reviewer and we streamlined our results to underline our main findings. In doing so, we also included a more extensive discussion of the biological implication of our findings for the alteration of proteostasis observed in pathologies such as neurodegeneration where mis-localization of a protein or alteration of its activity could be at the basis of changes in their turnover (Page 16, lines 611-642).

Reviewer 3 Report

The authors present an interesting study to address the influence of subcellular localization and protein functional state on protein turnover. The corresponding measurements have been conducted with care and the obtained results should have a good potential for meaningful conclusions. However, few points need to be addressed with more attention before publication.

1) Can the authors explain better why did they choose the three examples reported in the paper? There are several examples that can be taken into account, why did they prefer to report Rab5a, Calmodulin and Creatine kinase?

2) I'm wondering about the conclusion the authors reached regarding the influence of pI on protein stability. Studies of the thermodynamics
of folding and conformational stability are usually necessary for this kind of topic, otherwise it can be explored by a computational analysis. However, in the paper the theoretical isoelectric point, secondary structural features and the GRAVY score were computed for a small number of proteins and this cannot allow to make an adequate statistical correlation. I understand the authors wanted to present a complete picture regarding the influence of subcellular localization on protein turnover, but with these numbers both results and conclusions are really risky. My suggestion is to not press to much in the main text this topic that maybe can be moved to supplementary materials.

Author Response

Reviewer #3
The authors present an interesting study to address the influence of subcellular localization and protein functional state on protein turnover. The corresponding measurements have been conducted with care and the obtained results should have a good potential for meaningful conclusions.

We sincerely thank the reviewer for the overall positive assessment of our work.

However, few points need to be addressed with more attention before publication.
1) Can the authors explain better why did they choose the three examples reported in the paper? There are several examples that can be taken into account, why did they prefer to report Rab5a, Calmodulin and Creatine kinase?

We completely agree with the reviewer: there are infinite possibilities for choosing examples to explore. The reasoning behind the choice of these three particular examples is that we decided to include one kinase, a small GTPase and a protein modulated by the binding of a small ion and more
specifically Ca2+, which is essential for the regulation of cellular activity.
The reason for selecting a kinase is that mammalian genomes encode for ~500 kinases and ~100 kinase pseudogenes (Milanesi et al, 2005). Moreover, kinases are one of the most studied targets for pharmacological intervention and are a very-well defined family of enzymes, characterized
by a stereotypic catalytic domain and kinase-dead mutants are well described. CKB is one of the most abundant kinases in some organs including the brain, where protein turnover plays an essential
role.
The small GTPase Rab5 is a ubiquitous and very-well described small GTP-binding protein involved in endosomal recycling (Stenmark et al., 1994). GTPases, which include Rho, Ras and Rab proteins, are ‘molecular switches’ which get activated by GTP and inactivated by its hydrolysis to
GDP. This family of proteins, comprising overall >150 members is highly conserved across evolution and represents a ubiquitous regulatory mechanism in eukaryotic cells (Homma et al., 2020; Song et
al., 2019).
Ion binding proteins are also crucial regulators of cellular functions and Calmodulin is one of the most abundant proteins with this property, with extremely highly conserved structure across eukaryotes. Calmodulin has also the advantage of bearing an EF-hand motif, which has been
crystallized and coordinates calcium in a pentagonal bipyramidal configuration that is common to several calcium-binding proteins (Villalobo et al., 2019).
We have followed the suggestion of the reviewer and we have summarized some of the information reported here in the updated version of our manuscript (Page 12, lines 477-482 and Page 15, lines 581-600) to more extensively explain our choice.

2) I'm wondering about the conclusion the authors reached regarding the influence of pI on protein stability. Studies of the thermodynamics of folding and conformational stability are usually necessary for this kind of topic, otherwise it can be explored by a computational analysis. However, in the paper the theoretical isoelectric point, secondary structural features and the GRAVY score were computed for a small number of proteins and this cannot allow to make an adequate statistical correlation. I understand the authors wanted to present a complete picture regarding the influence of subcellular
localization on protein turnover, but with these numbers both results and conclusions are really risky. My suggestion is to not press to much in the main text this topic that maybe can be moved to supplementary materials

We are in complete agreement with the Reviewer. We aimed at providing a complete picture of our results, but, since this is actually superfluous for our main conclusions (and can be misleading as we noticed from the comments of Reviewer #2), we have moved this part to the supplement (Supp. Fig. 3), as suggested by the reviewer.

Round 2

Reviewer 1 Report

The authors have improved the manuscript and answered all the queries. 

Recommending for publication. 

Reviewer 2 Report

The manuscript has been significantly improved. I believe it is now suitable for publication.